# Whole-Exome Sequencing Reveals Migraine-Associated Novel Functional Variants in Arab Ancestry Females: A Pilot Study

**DOI:** 10.3390/brainsci12111429

**Published:** 2022-10-24

**Authors:** Johra Khan, Lubna Al Asoom, Ahmad Al Sunni, Nazish Rafique, Rabia Latif, Majed Alabdali, Azhar Alhariri, Majed Aloqaily, Sayed AbdulAzeez, Sadaf Jahan, Saeed Banawas, J. Francis Borgio

**Affiliations:** 1Department of Medical Laboratory Sciences, College of Applied Medical Sciences, Majmaah University, Majmaah 11952, Saudi Arabia; 2Health and Basic Sciences Research Center, Majmaah University, Majmaah 11952, Saudi Arabia; 3Department of Physiology, College of Medicine, Imam Abdulrahman Bin Faisal University, Dammam 31541, Saudi Arabia; 4Department of Neurology, College of Medicine, Imam Abdulrahman Bin Faisal University, Dammam 31952, Saudi Arabia; 5Department of Genetic Research, Institute for Research and Medical Consultations (IRMC), Imam Abdulrahman Bin Faisal University, Dammam 31441, Saudi Arabia; 6Department of Biomedical Sciences, Oregon State University, Corvallis, OR 97331, USA; 7Department of Epidemic Diseases Research, Institute for Research and Medical Consultations (IRMC), Imam Abdulrahman Bin Faisal University, Dammam 31441, Saudi Arabia

**Keywords:** exome, migraine, whole-exome sequencing, *RETNLB*, Saudi Arabia, next-generation sequencing, risk variants

## Abstract

Migraine, as the seventh most disabling neurological disease with 26.9% prevalence in Saudi females, lacks studies on identifying associated genes and pathways with migraines in the Arab population. This case control study aims to identify the migraine-associated novel genes and risk variants. More than 1900 Arab ancestry young female college students were screened: 103 fulfilled the ICHD-3 criteria for migraine and 20 cases confirmed in the neurology clinic were included for the study with age-matched healthy controls. DNA from blood samples were subjected to paired-end whole-exome sequencing. After quality control, 3365343 missense, frameshift, missense splice region variants and insertion–deletion (indels) polymorphisms were tested for association with migraine. Significant variants were validated using Sanger sequencing. A total of 17 (*p*-value 9.091 × 10^−05^) functional variants in 12 genes (*RETNLB*, *SCAI*, *ADH4*, *ESPL1*, *CPT2*, *FLG*, *PPP4R1*, *SERPINB5*, *ZNF66*, *ETAA1*, *EXO1* and *CPA6*) were associated with higher migraine risk, including a stop-gained frameshift (-13-14*SX) variant in the gene *RETNLB* (rs5851607; *p*-value 3.446 × 10^−06^). Gene analysis revealed that half of the significant novel migraine risk genes were expressed in the temporal lobe (*p*-value 0.0058) of the cerebral cortex. This is the first study exploring the migraine risk of 17 functional variants in 12 genes among Saudi female migraineurs of Arab ancestry using whole-exome sequencing. Half of the significant genes were expressed in the temporal lobe, which expands migraine pathophysiology and early identification using biomarkers for research possibilities on personalised genetics.

## 1. Introduction

Migraine is a multifactorial chronic neurological disorder with a global prevalence between 14 to 21.7% [1]. It creates a burden on the economy as well as on the normal lives of individuals [2]. It is considered the seventh most disabling disease around the world as it causes 2.9% of years of life lost to disability (YLD) [3,4,5]. In Saudi Arabia, the prevalence of migraines in females is around 26.9% and is considered higher than the global average [6,7]. In addition, recent studies in this population revealed that high oestrogen levels could be involved in mediating the non-menstrual-related migraine among young Saudi females [8], and serum ApoE was reported as an excellent diagnostic marker for the same subjects with migraine in ictal or interictal phase [9]. However, genetic studies are deficient concerning migraines in this population [8].

Migraine is classified mainly into two forms: migraine with aura (MA) and migraine without aura (MO). It can also be classified into a chronic and episodic migraine. Hemiplegic migraine is another type of MA and is a severe and rare condition that affects one side of the body and causes temporary numbness [9]. It once was believed that migraine is a vascular disease; however, many recent types of research confirmed the involvement of multiple mechanisms related to the brain structures and processes [10]. Furthermore, accumulated evidence demonstrates a genetic background for the transmission of the disease. Many epidemiological studies showed the passage of migraine in families with a concordance rate of 1- to 2-fold in monozygotic twins in comparison to dizygotic twins [11]. Different approaches were used to determine the genetic factors and the DNA variants that might be responsible for migraine. Some studies focused on candidate-gene-associated studies (CGAS) to identify specific genetic markers associated with migraine but the negative result of these studies and the development of cost-effective techniques resulted in genome-wide association studies [12]. In these studies, hundreds of thousands of single-nucleotide polymorphisms (SNPs) were screened for their association with migraine. Studies from headache clinics of the Netherlands, Germany and Finland on a 10,747 people led to the identification of rs835740, a single, significant SNP [13]. A similar recent study identified 38 different SNPs and 7 loci, *TRPM8*, *LRP1*, *FHL5*, *TSPAN2*, *ASTN2*, near *FGF6*, and *PHACTR1*, but none of them were found related to migraine without aura [14]. However, there are no detailed studies on migraine-associated genes from the Arab population [8]. Focused studies on SNPs especially in females are needed, as they are the most affected patients of migraine. Identification of these migraine-associated SNPs needs collaboration from different areas of the world to collect sufficient information from populations with different ethnicities and draw the full picture of the genetic predisposition of migraine. Therefore, this study was designed to identify the associated gene variants and single-nucleotide polymorphisms with migraine in young Saudi females using whole-exome sequencing analysis.

## 2. Materials and Methods

The study protocol was approved by the Institutional Review Board of Imam Abdulrahman Bin Faisal University (IRB number: IRB-2021-01-250) and was conducted following the Declaration of Helsinki. More than 1900 Arab ancestry young female college students were screened, 103 fulfilled the ICHD-3 criteria for migraine and 20 cases confirmed in the neurology clinic were included for the study with age-matched controls. This is an exome-wide association study involving 40 participants: 20 controls (healthy subjects) and 20 cases (migraineurs) conducted in the female campus of the College of Medicine and the Institute for Research and Medical Consultations (IRMC) of Imam Abdulrahman bin Faisal University (IAU), Dammam, Saudi Arabia. Subjects were recruited during the year 2021 by convenience sampling and were all Saudi female college students with an age range of 18–30. For the cases to be included in the study, they had to be diagnosed with migraine by a neurologist and satisfy the criteria of the International Classification of Headache Disorders 3rd edition (ICHD). Controls were healthy female subjects of the same age group and with no complaints of headache.

All participants filled a written informed consent for their enrolment in the study, and then they were interviewed and asked to fill an electronic datasheet (Appendix A). The datasheet included the demographic characteristics (age, marital status, college level, height, weight, BMI). In addition, it included specific questions related to migraine such as frequency of attacks/month, the severity of the attack (using the visual scale 1–10), associated symptoms, presence of triggers (stress, lack of sleep, missed meal or fasting, physical activity, noise, smell, strong lights, fluctuation of weather or temperature, food, relation to the menstrual cycle), type of migraine (with aura, without aura), family history of migraine, history (other chronic diseases) and use of medications (for migraine, for other diseases).

### 2.1. Whole-Exome Sequencing and Statistical Data Analysis

Blood samples were obtained from the study participants in EDTA-vacutainers. Deoxyribonucleic acid was extracted using QIAamp DNA Blood Mini Kit (Qiagen, Hilden, Germany), DNA purity was checked using nanodrop and concentration was determined using a qubit fluorometer. DNA integrity was tested using agarose gel electrophoresis. Paired-end whole-exome sequencing was conducted for all the samples and subjected to quality screening. The qualified DNA sample was randomly fragmented (150 to 200 bp) and adapters were ligated to both ends and purified by the AMPure beads to excise about 200 bp fragments. Fragmented DNA was subjected to ligation-mediated polymerase chain reaction, followed by the SureSelect Library for enrichment. Captured ligation-mediated polymerase chain reaction products were estimated using Bioanalyzer and loaded on Hiseq platform for high-throughput sequencing. Sample depth means ≥ 7.5, sample variant call rate ≥ 0.5 and sample genotype quality mean ≥ 28 were considered for good quality sample. The conditions such as Phred score quality ≥ 30, raw read depth ≥ 10 and mapping quality ≥ 30 were applied for filtering the good quality variants. The Hail standard (python package) was used for the entire genome-wide association study pipeline analysis as well as for the quality check of the samples, variants and genotypes. Manhattan plot and QQ plot of the association of SNPs with migraine as statistical significance in terms of *p*-values on a genomic scale were constructed. Gene-level linkage disequilibrium analysis of the SNPs in the significant gene was conducted using Haploview 4.2.

### 2.2. Statistical Analysis

Data were analysed using a statistical package for the social sciences (SPSS) software version 21. Student’s *t*-test was used for identifying the significant difference among the demographic characteristics between migraineurs and the controls. The most significant top 10 genes were analysed using the gene functional classification tool DAVID to identify the significance in the site of expression under the category of GNF_U133A_QUARTILE (*p*-value < 0.05) [15]. The functional annotation of the top 50 highly associated genes (with the lowest *p*-value < 0.00023) was performed using the Uniprot database. Further, the GO and pathway enrichment were carried out using enrichR server, and pathway involvement of the genes was performed by KEGG search and colour pathway server. All the significant markers identified from the exome-wide association study and genes were selected for the expression profile analysis in the brain and related tissues using DAVID. Brain and related tissue-expressed genes were separated and analysed for KEGG pathway enrichment.

## 3. Results

### 3.1. Study Population

Study participants (Table 1) were drawn from Arab ancestries. The demographic characteristics of the 40 selected Saudi Arabian subjects including 20 migraine patients and controls (*n* = 20) are presented in Table 1. The clinical characteristics and the frequency of the precipitating factors of headache attack in the migraineurs of the study are presented in Table 2.

### 3.2. Single-Variant Analysis

After quality controls for the variants were obtained in the whole-exome sequencing, 3,365,343 variants were satisfied for further exome-wide association analyses. Our study highlights the added influence of considering the functional variants such as missense variants, frameshift variant and missense splice region variant in the analysis: *p*-values < 0.00001 were set from the exome-wide association analysis (Table 3) to prioritise the top migraine-associated variants in Arab ancestry. The entire list of migraine-associated (*p*-value < 0.00001) variants identified through exome sequencing are shown in Appendix A. Seventeen variants were found to be the most significant (9.091 × 10^−5^) functional variants distributed among twelve genes (*RETNLB*, *SCAI*, *ADH4*, *ESPL1*, *CPT2*, *FLG*, *PPP4R1*, *SERPINB5*, *ZNF66*, *ETAA1*, *EXO1* and *CPA6*) (Table 3 and Figure 1). The stop-gained frameshift (-13-14*SX) variant in the gene *RETNLB* was found to be the most significant functional variant (rs5851607; *p*-value = 3.446 × 10^−06^).

### 3.3. Gene Analysis

The most significant top 12 genes were analysed using the gene functional classification tool DAVID to identify the significance in the site of expression. The analysis revealed that 6 out of 12 genes (Table 3) were significantly expressed in the temporal lobe (*p*-value = 0.00582) (Figure 2). The functional variants with a *p*-value between 9.091 × 10^−05^ to 0.05 on the genes identified as expressed in the temporal lobe revealed the significance of *FLG* gene with 37 functional variants (Table 4). The significant SNPs observed in the *FLG* gene, and their amino acid position, are presented in Figure 3. The gene-level linkage disequilibrium analysis (Haploview 4.2) of the 53 SNPs in the *FLG* gene reveals significant association for all these SNPs (Chi-Square = 8.556; *p*-value = 0.0034) (Figure 3). The most significant three markers haplotype rs3126075G, rs7532285T and rs7540123G (Chi-Square = 7.64; *p*-value = 0.0057) appear to associate significantly with migraine, while the opposite alleles rs3126075C, rs7532285C and rs7540123C (Chi-Square = 3.81; *p*-value = 0.0407) are the protective type of haplotype. The presence of variants at *FLG* was confirmed using Sanger sequencing with the designed primers (forward primer, FLGF: 5′ CCTCTACCAGGTGAGCACTCATGAACAGTCTG 3′, and reverse primer, FLGR: 5′ TCTCTGACTGCAGATGAAGCTTGTCCGTGCC 3′). Sanger sequencing revealed additional variants in the gene and the need to check their association with migraine (Figure 4).

### 3.4. Pathways Analysis

Gene ontology and pathway enrichment of the top 50 genes revealed that the migraine subjects are moderately significant for the organic hydroxy compound catabolic process (GO:1901616; *p*-value = 9.12893 × 10^−05^; adjusted *p*-value = 0.026108731) and quinone metabolic process (GO:1901661; *p*-value = 0.000272123; adjusted *p*-value = 0.038913602) (Table 5; Appendix A). All the significant markers having genes were checked for the expression profile in the brain-related tissues using DAVID. A total of 6305 genes were present for DAVID among 11958 genes presented for expression analysis using DAVID. A total of 1349 genes were separated based on the expression in the brain and related tissues were analysed for KEGG pathway enrichment; the results revealed that 34 genes were found to be significantly (term *p*-value = 8.071 × 10^−12^; adjusted *p*-value = 2.364 × 10^−09^) associated with systemic lupus erythematosus (Table 6; Appendix A). Furthermore, the pathways such as focal adhesion, ECM-receptor interaction, human papillomavirus infection, alcoholism, pathways in cancer, pi3k-akt signalling pathway and cholesterol metabolism are significantly associated with migraine in the Saudis (adjusted *p*-value ≤ 2.192 × 10^−05^) (Table 5).

## 4. Discussion

Genetics play a significant part in migraines, in addition to other factors [16]. However, migraine genetic predisposition does not follow a direct Mendelian pattern. The common form of migraine is most probably polygenic and involves multiple variants at several genetic loci that possibly interact with multiple environmental factors. Exome-wide association study is the most successful method to identify the genes involved in a disease. In this methodology, cohorts of migraine cases and controls are explored for any differences in allele frequencies of single-nucleotide polymorphisms (SNPs) to identify genetic risk factors. There is no single genetic variant that can explain migraine heterogeneity across populations. We performed the first exome-wide association study of migraine in Arab ancestry. Through exome sequencing, we identified an entire list of migraine-associated (*p*-value < 0.00001) variants and prioritised 17 as the most significant (9.091 × 10^−05^) functional variants distributed among 12 genes (*RETNLB*, *SCAI*, *ADH4*, *ESPL1*, *CPT2*, *FLG*, *PPP4R1*, *SERPINB5*, *ZNF66*, *ETAA1*, *EXO1* and *CPA6*) in the Saudi females suffering from migraine. All of these were novel and have not been documented in earlier studies involving other populations such as Europeans [17] and Chinese [18].

The stop-gained frameshift variant in the gene *RETNLB* is the most significant functional variant, rs5851607; this gene encodes a bactericidal protein, Resistin-like molecule β (RELMβ), which is released from colonic cells to destroy Gram-negative bacteria. Migraine may be associated with diseases such as irritable bowel syndrome (IBS), inflammatory bowel syndrome, and celiac disease [19]. The *ADH4* gene encodes the alcohol dehydrogenase enzyme and variations in this gene are associated with alcohol dependence [20]. The *CPA6* gene encodes the Carboxypeptidase A6 enzyme, and its mutations can predispose to various types of epilepsy [21]. As shown in Figure 2 and Table 3, a total of 36 functional variants were found to be significant in the gene *FLG.* This gene encodes a protein called profilaggrin present in the epidermis of the skin. This protein is important for the skin’s barrier function. Functional variations in this gene can cause sensitisation or atopic dermatitis [22] and might be the underlying mechanism of migraine-associated allodynia [23]. The *ETAA1* gene encodes the protein Ewing tumour-associated antigen 1. This protein functions as a DNA replication stress response protein [24]. The *CPT2* gene encodes the carnitine palmitoyltransferase 2 enzyme, which is essential for fatty acid oxidation. The *ESPL1* gene codes a protease separase/separin which causes separation of sister chromatids in mitosis [25]. The *SERPINB5* gene encodes a protein Maspin, a tumour suppressor that binds directly to extracellular matrix components and inhibits tumour-induced angiogenesis, invasion and metastatic spread [26]. SCAI encodes a protein that suppresses cancer cell invasion [27]. The significant association of multiple genes to migraine might help to explain the wide spectrum of migraine phenotypes.

The morphological changes in the temporal lobe were reported to be associated with migraines [27]. Recently, a reduction in grey matter volume in the temporal lobe was observed in migraine patients [28]. Gene analysis revealed that 6 of the significant 12 novel migraine risk genes were expressed in the temporal lobe of the cerebral cortex. The present study adds a molecular insight into the observations on the temporal lobe [29] and migraine-associated genes [30,31] and opens new avenues for migraine research. The current study will help in power calculations in the future and will provide potential loci to look for in replication studies. This may facilitate a thorough understanding of migraine pathophysiology and its underlying molecular mechanism, and open avenues for more precise diagnosis and therapeutic strategies targeting migraine patients of Arab ancestry. The study may also help in the polygenic risk scoring of the patients. The number of study subjects is one of the notable limitations in the study.

## 5. Conclusions

Our study is the first one exploring migraine genetic variations in Arab ancestry. Seventeen significant functional variants including a stop gained in twelve genes are the migraine risk variants in Arab ancestry. Half of the significant novel migraine risk genes are expressed in the temporal lobe of the brain.

## Figures and Tables

**Figure 1 brainsci-12-01429-f001:**
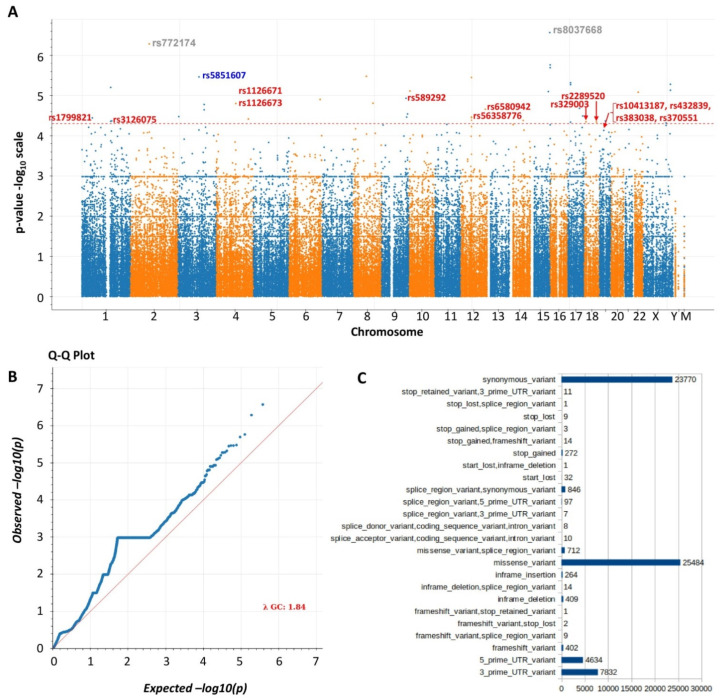
Manhattan plot (**A**) and QQ plot (**B**) of the association of SNPs with migraine as statistical significance in terms of *p*-values on a genomic scale. SNP numbers with blue colour indicate the highly associated (*p* < 3.44623 × 10^−06^) stop-gained frameshift variant. SNP numbers with red colour indicate the highly associated (*p* < 0.00001) missense variants. SNP numbers with ash colour indicate the highly associated (*p* < 0.00001) intronic and 5 prime variants. Chr15: rs8037668 [‘T’, ‘C’] (*p*-value 2.7135 × 10^−07^), LOC400464 intron variant, non-coding transcript variant; and Chr 2: rs772174 [‘A’, ‘G’], (*p*-value 5.2006 × 10^−07^), ITPRIPL1 protein-coding 5 prime UTR variant showed the highest *p*-value. (**C**) Consequences of coding variants. Full list of variants (*p*-value < 0.00001) is listed in Appendix A.

**Figure 2 brainsci-12-01429-f002:**
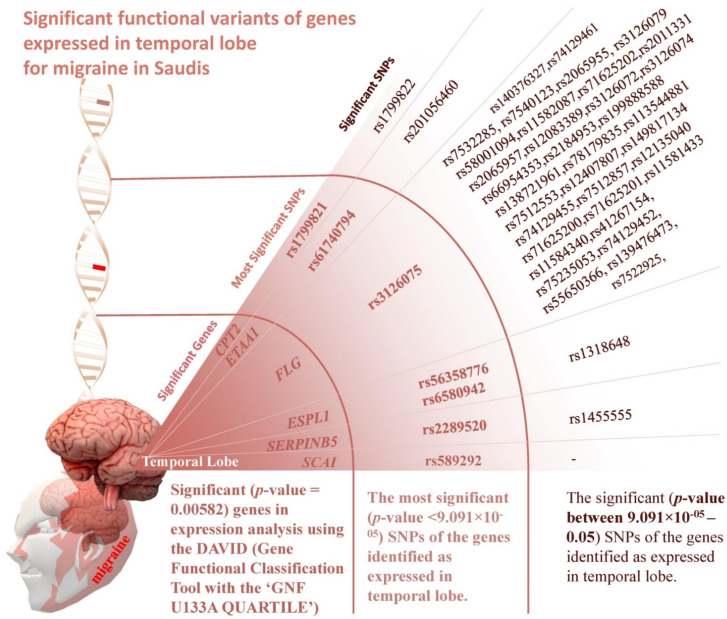
Significant functional variants of genes expressed in the temporal lobe (*p*-value = 0.00582) for migraine in Saudis. The most significant genes (top 12 as listed in Table 3) associated functionally were used as input to identify the expression nature of them using the gene functional classification tool DAVID with the ‘GNF U133A QUARTILE’. The most significant (*p*-value < 9.091 × 10^−05^) functional variants are presented in the middle path. The functional variants with a *p*-value between 9.091 × 10^−05^ to 0.05 on the genes identified as expressed in the temporal lobe are also presented. A total of 37 functional variants are found to be significant in the gene *FLG* (full list presented in Table 4).

**Figure 3 brainsci-12-01429-f003:**
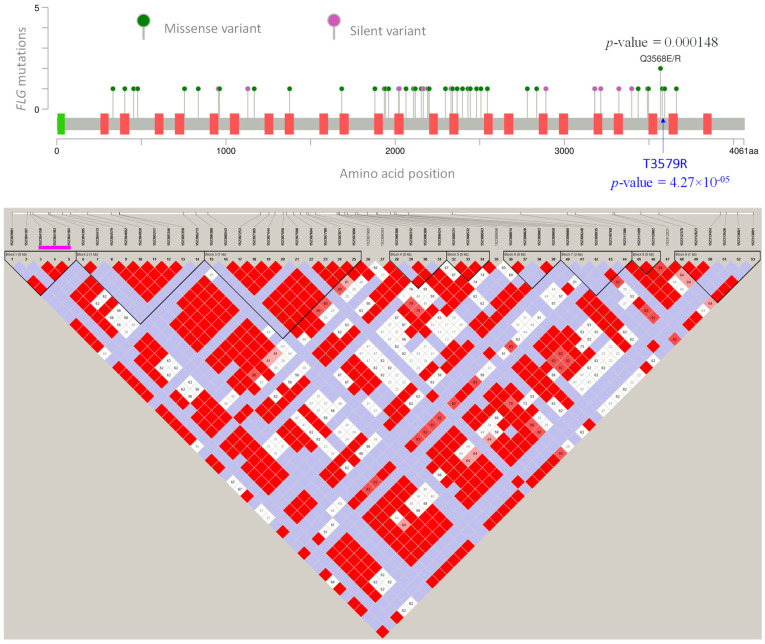
Top: variants observed in *FLG* gene. Pink lollipop indicates silent mutation; green lollipop indicates missense variations. Amino acid substitution in blue colour indicates the most significant variant. Box coloured green indicates the calcium-binding domain. Bottom: Haplomap of gene-level linkage disequilibrium analysis of the SNPs in the *FLG* gene. Pink line: the most significant three markers rs3126075G, rs7532285T and rs7540123G.

**Figure 4 brainsci-12-01429-f004:**
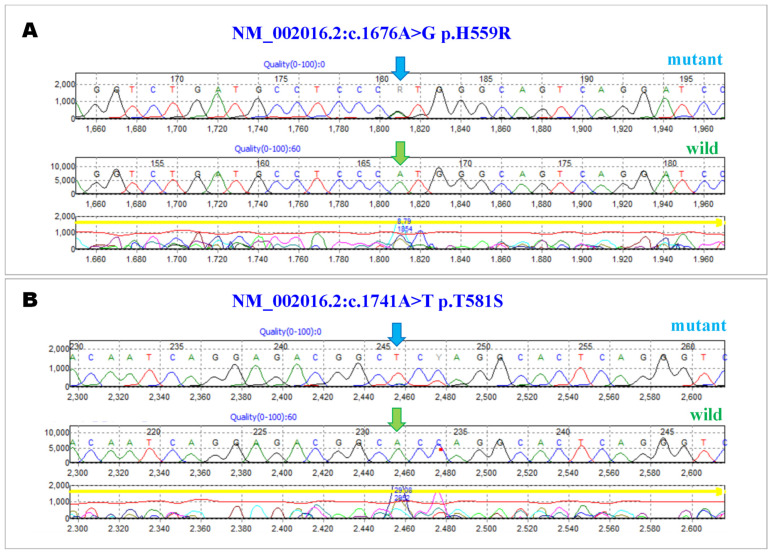
Electropherogram of *FLG* gene with (**A**) NM_002016.2:c.1676A>G p.H559R and (**B**) NM_002016.2:c.1741A>T p.T581S. The blue arrow indicates mutant. The green arrow indicates wild type.

**Table 1 brainsci-12-01429-t001:** Demographic characteristics of the migraineurs and the controls.

	Control *n* = 20	Migraineurs *n* = 20	*p*-Value *
Age (years)	21.86 ± 1.75	22.10 ± 3.63	0.818
Bodyweight (kg)	56.21 ± 14.02	63.00 ± 12.58	0.155
BMI	21.48 ± 5.39	24.66 ± 5.18	0.095

*** Student’s *t*-test.

**Table 2 brainsci-12-01429-t002:** Clinical characteristics of the migraineurs of the study and the frequency of the precipitating factors.

Variable	Description	
Type of migraine	With aura 11 (57.9%), without aura 9 (42.1%)	
Family history	Yes 11 (55%), no 9 (45%)	
Use of medications	No medication 7 (35%), pain killer 10 (50%), prophylaxis 3 (15%)	
Number of attacks/months	Ranges from 2–28 attacks/month with 2 attacks/month the most common	
Duration of the attack	Ranges from 4 h to >72 h with the most common 12–24 h	
The severity of the attack (Visual scale 1–10)	Ranges from 7–10	
	**Precipitating factors**	
	**Yes**	**No**
Sleep disturbances	17 (85%)	3 (15%)
Stress	16 (80%)	4 (20%)
Bright light	18 (90%)	2 (10%)
Excessive noise	15 (75%)	5 (25%)
Strong smells	12 (60%)	8 (40%)
Weather changes	8 (40%)	12 (60%)
Skipped meal	10 (50%)	10 (50%)
Physical exertion	5 (25%)	15 (75%)
Certain types of food	5 (25%)	15 (75%)
Coffee	2 (10%)	18 (90%)

**Table 3 brainsci-12-01429-t003:** List of migraine-associated (*p*-value < 0.00001) exonic functional variants identified through exome sequencing.

	Locus. Contig	Variation ID	Gene SYMBOL	MAF	Alleles	Amino Acids	Codons	*p*-Value
1	chr3	rs5851607 *	*RETNLB*	0.18	[‘G’, ‘GGGGGATTA’]	-13-14*SX	-/TAATCCCC	3.446 × 10^−06^
2	chr9	rs589292	*SCAI*	0.2	[‘C’, ‘T’]	A37T	Gct/Act	1.169 × 10^−05^
3	chr4	rs1126671	*ADH4*	0.26	[‘T’, ‘C’]	I309V	Att/Gtt	1.575 × 10^−05^
4	chr4	rs1126673 ^$^	*ADH4*	0.26	[‘C’, ‘T’]	V393I	Gtc/Atc	1.575 × 10^−05^
5	chr12	rs6580942	*ESPL1*	0.3	[‘C’, ‘A’]	A25D	gCc/gAc	3.413 × 10^−05^
6	chr1	rs1799821	*CPT2*	0.28	[‘G’, ‘A’]	V368I	Gtc/Atc	3.585 × 10^−05^
7	chr1	rs3126075	*FLG*	0.23	[‘G’, ‘C’]	T3579R	aCg/aGg	4.266 × 10^−05^
8	chr18	rs329003	*PPP4R1*	0.28	[‘T’, ‘C’]	I381V	Ata/Gta	4.500 × 10^−05^
9	chr18	rs2289520	*SERPINB5*	0.14	[‘G’, ‘C’]	V187L	Gtc/Ctc	4.556 × 10^−05^
10	chr12	rs56358776	*ESPL1*	0.26	[‘G’, ‘A’]	R1561Q	cGg/cAg	5.923 × 10^−05^
11	chr19	rs10413187	*ZNF66*	0.14	[‘C’, ‘A’]	Q66K	Cag/Aag	7.328 × 10^−05^
12	chr19	rs432839	*ZNF66*	0.22	[‘G’, ‘T’]	C173F	tGc/tTc	7.320 × 10^−05^
13	chr19	rs383038	*ZNF66*	0.14	[‘T’, ‘C’]	F188L	Ttt/Ctt	7.328 × 10^−05^
14	chr19	rs370551	*ZNF66*	0.14	[‘A’, ‘G’]	T420A	Act/Gct	7.328 × 10^−05^
15	chr2	rs61740794	*ETAA1*	0.4	[‘G’, ‘A’]	E673K	Gaa/Aaa	8.508 × 10^−05^
16	chr1	rs735943	*EXO1*	0.42	[‘A’, ‘G’]	H354R	cAt/cGt	8.978 × 10^−05^
17	chr8	rs17343819 ^$^	*CPA6*	0.22	[‘T’, ‘C’]	N249S	aAt/aGt	9.091 × 10^−05^

* Stop-gained frameshift variant; ^$^ missense splice region variant. Significant (*p*-value < 0.00001) functional and other variants identified through exome sequencing are listed in Appendix A. MAF: minor allele frequency.

**Table 4 brainsci-12-01429-t004:** List of significant migraine-associated missense variants of *FLG* gene in chromosome 1.

	SNP	Alleles	MAF	Amino Acids	Protein Position	Codons	*p*-Value
1	rs3126075	[‘G’, ‘C’]	0.23	T/R	3579	aCg/aGg	4.27 × 10^−05^
2	rs7532285	[‘T’, ‘C’]	0.05	Q/R	3568	cAg/cGg	0.000148
3	rs7540123	[‘G’, ‘C’]	0.05	Q/E	3568	Cag/Gag	0.000148
4	rs2065955	[‘C’, ‘G’]	0.3	G/A	3436	gGa/gCa	0.000592
5	rs3126079	[‘G’, ‘T’]	0.3	H/Q	1961	caC/caA	0.000592
6	rs58001094	[‘G’, ‘C’]	0.3	A/G	1167	gCa/gGa	0.000592
7	rs11582087	[‘T’, ‘G’]	0.04	S/R	2836	Agt/Cgt	0.00092
8	rs71625202	[‘C’, ‘G’]	0.08	S/T	2366	aGt/aCt	0.00092
9	rs139476473	[‘C’, ‘T’]	0.04	D/N	2339	Gac/Aac	0.00092
10	rs2065957	[‘A’, ‘C’]	0.17	V/G	3179	gTg/gGg	0.001166
11	rs12083389	[‘C’, ‘G’]	0.15	E/D	3593	gaG/gaC	0.002327
12	rs3126072	[‘C’, ‘T’]	0.22	G/R	2545	Gga/Aga	0.003517
13	rs3126074	[‘G’, ‘C’]	0.22	H/Q	2507	caC/caG	0.003517
14	rs2011331	[‘T’, ‘C’]	0.22	T/A	454	Aca/Gca	0.003517
15	rs66954353	[‘T’, ‘G’]	0.13	K/Q	2192	Aaa/Caa	0.003521
16	rs2184953	[‘A’, ‘G’]	0.32	Y/H	2194	Tat/Cat	0.004922
17	rs140376327	[‘G’, ‘A’]	0.04	R/W	2430	Cgg/Tgg	0.010229
18	rs12135040	[‘C’, ‘G’]	0.04	G/R	1936	Ggg/Cgg	0.010229
19	rs138721961	[‘C’, ‘T’]	0.04	R/H	402	cGc/cAc	0.010229
20	rs78179835	[‘C’, ‘G’]	0.08	E/D	2297	gaG/gaC	0.014364
21	rs113544881	[‘A’, ‘T’]	0.07	L/H	1943	cTt/cAt	0.016921
22	rs74129452	[‘T’, ‘G’]	0.18	Q/H	2154	caA/caC	0.018862
23	rs7512553	[‘A’, ‘G’]	0.18	Y/H	2119	Tat/Cat	0.018862
24	rs7522925	[‘G’, ‘A’]	0.18	A/V	2108	gCg/gTg	0.018862
25	rs7512857	[‘A’, ‘C’]	0.18	S/A	2020	Tca/Gca	0.018862
26	rs12407807	[‘C’, ‘T’]	0.16	R/H	1684	cGc/cAc	0.023383
27	rs75235053	[‘C’, ‘G’]	0.06	S/T	3662	aGt/aCt	0.026472
28	rs199888588	[‘A’, ‘G’]	0.1	W/R	962	Tgg/Cgg	0.027008
29	rs74129455	[‘T’, ‘G’]	0.1	K/Q	2064	Aaa/Caa	0.031164
30	rs149817134	[‘G’, ‘T’]	0.04	H/N	1880	Cac/Aac	0.03179
31	rs55650366	[‘A’, ‘G’]	0.14	L/S	2481	tTg/tCg	0.035772
32	rs71625200	[‘T’, ‘C’]	0.14	K/E	2444	Aag/Gag	0.035772
33	rs71625201	[‘C’, ‘G’]	0.14	E/Q	2398	Gag/Cag	0.035772
34	rs11581433	[‘T’, ‘C’]	0.14	R/G	1376	Aga/Gga	0.035772
35	rs74129461	[‘C’, ‘T’]	0.14	E/K	755	Gaa/Aaa	0.035772
36	rs11584340	[‘G’, ‘A’]	0.14	P/S	478	Cct/Tct	0.035772
37	rs41267154	[‘C’, ‘A’]	0.14	G/V	332	gGc/gTc	0.035772

MAF: minor allele frequency.

**Table 5 brainsci-12-01429-t005:** KEGG pathway enrichment from the top 50 genes associated with the exome wide association study analysis.

Term	*p*-Value	Number of Genes from Top 50 List	Total Genes Involved	Genes from the Top 50 List
p53 signalling pathway	0.000776571	3 *	72	TNFRSF10B; SERPINB5; ATR
Fatty acid degradation	0.005418766	2 ^$^	44	ADH4; CPT2
Metabolism of xenobiotics by cytochrome P450	0.014752518	2	74	ADH4; AKR1C1
Cell cycle	0.038497072	2	124	ESPL1; ATR
Mannose type O-glycan biosynthesis	0.055975781	1	23	FKRP
Mismatch repair	0.055975781	1	23	EXO1
Tyrosine metabolism	0.086243722	1	36	ADH4
Human T-cell leukaemia virus 1 infection	0.10395435	2	219	ESPL1; ATR
Vasopressin-regulated water reabsorption	0.104395242	1	44	AQP4
Fanconi anaemia pathway	0.126588324	1	54	ATR
Steroid hormone biosynthesis	0.139644523	1	60	AKR1C1
Cytosolic DNA-sensing pathway	0.146100703	1	63	CCL4L2
Retinol metabolism	0.154635136	1	67	ADH4
Glycolysis/Gluconeogenesis	0.156755649	1	68	ADH4

* The p53 signalling pathway with the significant genes is presented in Appendix A. ^$^ The fatty acid degradation with the significant genes is presented in Appendix A.

**Table 6 brainsci-12-01429-t006:** KEGG pathway enrichment from the 1349 genes based on the expression in brain-related tissues associated with the exome-wide association analysis.

Term	*p*-Value	Adjusted *p*-Value	Genes from 1349 List *	Total Genes
Systemic lupus erythematosus	8.071 × 10^−12^	2.364 × 10^−09^	34	133
Focal adhesion	2.665 × 10^−11^	3.905 × 10^−09^	42	199
ECM-receptor interaction	4.693 × 10^−10^	4.583 × 10^−08^	24	82
Human papillomavirus infection	3.167 × 10^−09^	2.319 × 10^−07^	53	330
Alcoholism	1.142 × 10^−08^	6.693 × 10^−07^	35	180
Pathways in cancer	2.191 × 10^−08^	1.069 × 10^−06^	71	530
PI3K-Akt signalling pathway	9.237 × 10^−08^	3.866 × 10^−06^	52	354
Cholesterol metabolism	5.986 × 10^−07^	2.192 × 10^−05^	15	50

* The list of significant genes is presented in Appendix A.

## Data Availability

All data will be available on reasonable request from the corresponding author.

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
