# Peer review of "Whole-Exome Sequencing Reveals Migraine-Associated Novel Functional Variants in Arab Ancestry Females: A Pilot Study"

_brainsci, 2022, doi:10.3390/brainsci12111429_

Round 1

Reviewer 1 Report

Title: Whole-exome sequencing reveals migraine associated novel functional variants in Arab ancestry females: A pilot study

In this work more than 1900 Arab ancestry young female college students were screened, 103 were fulfilled the ICHD-3 criteria for migraine, while 20 cases confirmed in the neurology clinic were included for the study with age matched healthy controls. DNA from blood samples were subjected for paired-end whole-exome sequencing. After quality control, 3365343 missense, frameshift, mis- sense splice region variants and insertion-deletion (indels) polymorphisms were tested for associ-ation with migraine. Significant variants were validated using Sanger sequencing. Seventeen 28
(p-value 9.091×10-05) functional variants in 12 genes (RETNLB, SCAI, ADH4, ESPL1, CPT2, FLG, PPP4R1, SERPINB5, ZNF66, ETAA1, EXO1 and CPA6) were associated with higher migraine risk including a stop gained frameshift (-13-14*SX) variant in the gene RETNLB (rs5851607; p-value 31
3.446×10-06). Gene analysis revealed that half of the significant novel migraine risk genes expressed in the temporal lobe (p-value 0.0058) of the cerebral cortex. The authors claim that this is the first study exploring the migraine risk 17 functional variants in 12 genes among Saudi female migraineurs of Arab ancestry using whole-exome sequencing. Half of the significant genes were expressed in the temporal lobe, which expands migraine pathophysiology and early identification using biomarkers for research possibilities on personalised genetics.

General comment: Although the topic of this work is interesting the manuscript should be revised to increase its quality and impact.

Major points:

lines: “It is an exome wide association study involving 40 participants: 20 controls (healthy subjects) and 20 cases (migraineurs) conducted in the female campus of the College of Medicine and the Institute for Research and Medical Consultations (IRMC) of Imam Abdulrahman bin Faisal University (IAU), Dammam, Saudi Arabia.”

1) A big limitation of the value of this work is the small number of participants. Therefore the authors should clearly explain what could be the statistical significance of this analysis related to a so small group of people and the resulting limitations to their finding.

Lines: “2.2. Statistical Analysis 119

Data were analysed using a statistical package for the social sciences (SPSS) software 120

version 21. The most significant top 10 genes were analysed using the gene functional 121

classification tool DAVID to identify the significance in the site of expression [15]. The 122

functional annotation of the top 50 highly associated genes (with the lowest p-value 123

<0.00023), was performed using the Uniprot database. Further, the GO and Pathway en- 124

richment was carried using enrichR server and pathway involvement of the genes were 125

done by KEGG search and colour pathway server. All the significant markers identified 126

from the exome wide association study and genes were selected for the expression profile 127

analysis in the brain and related tissues using DAVID. Brain and related tissue expressed 128

genes were separated and analysed for KEGG pathway enrichment. “

*) The authors should clearly explain the statistical methods used in this work. The reference to the SPSS is not enough. In addition, they should explain in a detailed way what the classification tool “DAVID” is and how it works.

Lines: “3.1. Study Population 131

Study participants (Table 1) were drawn from Arab ancestries. The demographic 132

characteristics of the 40 selected Saudi Arabian subjects including 20 migraine patients 133

and controls (n=20) are presented in Table 1. The clinical characteristics and the fre- 134

quency of the precipitating factors of headache attack in the migraineurs of the study are 135

presented in table 2”

and

Table 1: Demographic characteristics of the migraineurs and the controls.

Table 2: Clinical characteristics of the migraineurs of the study and the frequency of the precipitating factors:

*) The authors should explain whether these are results. They should be inserted in the “Methods” section.

Lines: “Figure 1: Manhattan plot (A) & QQ plot (B) of the association of SNPs with migraine as statistical significance in terms of p-values on a genomic scale. SNP numbers with blue colour indicate the highly associated (p<3.44623×10-06) stop gained frameshift variant. SNP numbers with red colour indicate the highly associated (p<0.00001) missense variants. SNP 1numbers with ash colour indicate the highly associated (p<0.00001) intronic and 5 prime variants. Chr15: rs8037668 ['T', C'] (p-value 2.7135×10-07) LOC400464 intron variant, non-coding transcript variant; and Chr 2: rs772174 ['A', 'G’] , (p-value 5.2006×10-07), ITPRIPL1 protein-coding 5 prime UTR variant showed the highest p-value. C: Consequences of coding variants. Full list of variants (p-value <0.00001) are listed in Table S1.

*) This figure should be provided in high resolution and should be improved to allow the interested readers to better explain all this research. The value of subfigures A, B and C should be explicitly presented to the readers. All the labels should be improved to be at least readable and understandable to the readers.

Paragraph “3.3. Genes Analysis “

*) This paragraph is crucial to the main aim of the work. However, it is quite complex and not clear to the readers. Please improve.

Figure 2+ Lines: “Figure 2: Significant functional variants of genes expressed in the temporal lobe (p-value = 0.00582) for migraine in Sau- 188

dis. The most significant genes (top 12 as listed in table 3) associated functionally were used as input to identify the ex- 189

pression nature of them using the gene functional classification tool, DAVID with the ‘GNF U133A QUARTILE’. The 190

most significant (p-value <9.091×10-05) functional variants are presented in the middle path. The functional variants with a 191

p-value between 9.091×10-05 to 0.05 on the genes identified as expressed in the temporal lobe are also presented. A total of 192

37 functional variants are found to be significant in the gene, FLG (Full list presented in Table 4).

*) Figure 2 may be improved. Please improve also the figure caption.

Figure 3: Top: Variants observed in FLG gene. Pink lollipop indicates silent mutation; Green lollipop indicates missense variations. Amino acid substitution in blue colour indicates the most significant variant. Box coloured green indicates the calcium-binding domain. Bottom: Haplomap of gene-level linkage disequilibrium analysis of the SNPs in the FLG gene. 197

Pink line: The most significant three markers rs3126075G, rs7532285T and rs7540123G.

*) Figure 3 is not clear. Please rework and improve the quality of presentation also by increasing the size of all the fonts used in the labels.

Figure 4: Electropherogram of FLG gene with (A): NM_002016.2:c.1676A>G p.H559R and (B): NM_002016.2:c.1741A>T 202

p.T581S. The Blue arrow indicates mutant. The green arrow indicates wild

*) See the previous comment.

Section “4. Discussion “

*) The section “Conclusions” seems to be lacking. The authors should better present to overall value of their work, since it is not clear how this work could “open new opportunities for migraine pathophysiology and genetic research”. Indeed, the main big question about the overall statistical significance of the work still remain.

Author Response

1) A big limitation of the value of this work is the small number of participants. Therefore the authors should clearly explain what could be the statistical significance of this analysis related to a so small group of people and the resulting limitations to their finding.

Lines: “2.2. Statistical Analysis 119

Data were analysed using a statistical package for the social sciences (SPSS) software 120

version 21. The most significant top 10 genes were analysed using the gene functional 121

classification tool DAVID to identify the significance in the site of expression [15]. The 122

functional annotation of the top 50 highly associated genes (with the lowest p-value 123

<0.00023), was performed using the Uniprot database. Further, the GO and Pathway en- 124

richment was carried using enrichR server and pathway involvement of the genes were 125

done by KEGG search and colour pathway server. All the significant markers identified 126

from the exome wide association study and genes were selected for the expression profile 127

analysis in the brain and related tissues using DAVID. Brain and related tissue expressed 128

genes were separated and analysed for KEGG pathway enrichment. “

*) The authors should clearly explain the statistical methods used in this work. The reference to the SPSS is not enough. In addition, they should explain in a detailed way what the classification tool “DAVID” is and how it works.

The study was conducted during the pandemic (COVID-19) due to which the number of participants was limited. In the statistical analysis section, it is mentioned that SPSS version 21 is used for p-value calculation and DAVID is a classification tool used for identifying the significance of the site of expression.  

Lines: “3.1. Study Population 131

Study participants (Table 1) were drawn from Arab ancestries. The demographic 132

characteristics of the 40 selected Saudi Arabian subjects including 20 migraine patients 133

and controls (n=20) are presented in Table 1. The clinical characteristics and the fre- 134

quency of the precipitating factors of headache attack in the migraineurs of the study are 135

presented in table 2”

and

Table 1: Demographic characteristics of the migraineurs and the controls.

Table 2: Clinical characteristics of the migraineurs of the study and the frequency of the precipitating factors:

*) The authors should explain whether these are results. They should be inserted in the “Methods” section.

These are part of the result.

Lines: “Figure 1: Manhattan plot (A) & QQ plot (B) of the association of SNPs with migraine as statistical significance in terms of p-values on a genomic scale. SNP numbers with blue colour indicate the highly associated (p<3.44623×10-06) stop gained frameshift variant. SNP numbers with red colour indicate the highly associated (p<0.00001) missense variants. SNP 1numbers with ash colour indicate the highly associated (p<0.00001) intronic and 5 prime variants. Chr15: rs8037668 ['T', C'] (p-value 2.7135×10-07) LOC400464 intron variant, non-coding transcript variant; and Chr 2: rs772174 ['A', 'G’] , (p-value 5.2006×10-07), ITPRIPL1 protein-coding 5 prime UTR variant showed the highest p-value. C: Consequences of coding variants. Full list of variants (p-value <0.00001) are listed in Table S1.

*) This figure should be provided in high resolution and should be improved to allow the interested readers to better explain all this research. The value of subfigures A, B and C should be explicitly presented to the readers. All the labels should be improved to be at least readable and understandable to the readers.

Figure 1 and  Figure 2 are replaced with high-resolution figures.

Paragraph “3.3. Genes Analysis “

*) This paragraph is crucial to the main aim of the work. However, it is quite complex and not clear to the readers. Please improve.

This paragraph explains the results of Figure 2, Figure 3 and Table 4.   

Figure 2+ Lines: “Figure 2: Significant functional variants of genes expressed in the temporal lobe (p-value = 0.00582) for migraine in Sau- 188

dis. The most significant genes (top 12 as listed in table 3) associated functionally were used as input to identify the ex- 189

pression nature of them using the gene functional classification tool, DAVID with the ‘GNF U133A QUARTILE’. The 190

most significant (p-value <9.091×10-05) functional variants are presented in the middle path. The functional variants with a 191

p-value between 9.091×10-05 to 0.05 on the genes identified as expressed in the temporal lobe are also presented. A total of 192

37 functional variants are found to be significant in the gene, FLG (Full list presented in Table 4).

*) Figure 2 may be improved. Please improve also the figure caption.

Figure 2 replaced and resubmitted

Figure 3: Top: Variants observed in FLG gene. Pink lollipop indicates silent mutation; Green lollipop indicates missense variations. Amino acid substitution in blue colour indicates the most significant variant. Box coloured green indicates the calcium-binding domain. Bottom: Haplomap of gene-level linkage disequilibrium analysis of the SNPs in the FLG gene. 197

Pink line: The most significant three markers rs3126075G, rs7532285T and rs7540123G.

*) Figure 3 is not clear. Please rework and improve the quality of presentation also by increasing the size of all the fonts used in the labels.

Improved

Figure 4: Electropherogram of FLG gene with (A): NM_002016.2:c.1676A>G p.H559R and (B): NM_002016.2:c.1741A>T 202

p.T581S. The Blue arrow indicates mutant. The green arrow indicates wild

*) See the previous comment.

Section “4. Discussion “

*) The section “Conclusions” seems to be lacking. The authors should better present to overall value of their work, since it is not clear how this work could “open new opportunities for migraine pathophysiology and genetic research”. Indeed, the main big question about the overall statistical significance of the work still remain.

The conclusion section is separated and the significance of the study was mentioned.

Reviewer 2 Report

Authors presented a very interesting study that helps to shed light on migraine genetic correlates and predisposition. This subject is still not clearly defined and authors work is very helpful for the community of headache researchers. Data presentation is very clear, with detailed tables and figures. 

I have few major concerns about methods and a minor concern about results. 

About methods: - why authors included just female patients/controls? There may be any gender selection bias; please, clarify or discuss. How were the 20 patients with a neurology clinic diagnosis selected from the 103 migraineurs sample? I would add the distinction between chronic and episodic migraineurs, to show in the results the two subpopulation; moreover, a sub analysis comparing exonic functional variants between the two groups could be interesting if the sample allow it.  

Among the results: I think there is no need to distinguish trigger factors in a "Yes/No" table, authors can just report the number of patients with the "Yes" triggers. 

The rest of the paper is fine. 

Author Response

About methods: - why authors included just female patients/controls? There may be any gender selection bias; please, clarify or discuss. How were the 20 patients with a neurology clinic diagnosis selected from the 103 migraineurs sample? I would add the distinction between chronic and episodic migraineurs, to show in the results the two subpopulations; moreover, a sub-analysis comparing exonic functional variants between the two groups could be interesting if the sample allows it.  

The reason for the selection of female patients is due to the large number of female Out Patients in comparison to males which is very low.  Clinical characteristics and the frequency of the precipitating factors of headache attacks in migraineurs were the reason for the selection of 20 patients out of 103. Due to the small sample size, it was not found suitable for further classification. 

I am thankful to the reviewer to provide the comments.

Among the results: I think there is no need to distinguish trigger factors in a "Yes/No" table, authors can just report the number of patients with the "Yes" triggers. 

As the number of trigger factors is large and not constant, it's necessary to distinguish between Yes and No.

Reviewer 3 Report

The current study uses WES to analyze the gene variants in Saudi females with migraine. I have a few comments below:

1. There are only 20 participants in each group in this study. However, there are various precipitating factors among the subjects. I wonder how representative the data are. Moreover, are the participants in their menstrual period when collected blood? Since estrogen is involved in migraine, is the estradiol level comparable among participants?

2. In the Methods part, please add information about the exome library preparation, sequencing platform, etc.

3. Figure 1A, 1B, 3 (haplomap), axis labels are not clear.

4. Any references of how those genes expressed in the temporal lobe are associated with migraine?

5. Again, how those enriched KEGG pathways are related with migraine?

Author Response

  1. There are only 20 participants in each group in this study. However, there are various precipitating factors among the subjects. I wonder how representative the data are. Moreover, are the participants in their menstrual period when collected blood? Since estrogen is involved in migraine, is the estradiol level comparable among participants?

The study was conducted during the pandemic (COVID-19) because of fewer participant registration. No blood samples were collected during the menstrual period. 

2. In the Methods part, please add information about the exome library preparation, sequencing platform, etc.

Section added and highlighted in the method section.  

3. Figure 1A, 1B, 3 (haplomap), axis labels are not clear.

Figure 1 s replaced with the clear figure.

4. Any references of how those genes expressed in the temporal lobe are associated with migraine?

Reference added.

5. Again, how those enriched KEGG pathways are related with migraine?

We selected migraine-associated genes from this study for KEGG pathway enrichment.

Round 2

Reviewer 1 Report

Title: Whole-exome sequencing reveals migraine associated novel functional variants in Arab ancestry females: A pilot study

In this work more than 1900 Arab ancestry young female college students were screened, 103 were fulfilled the ICHD-3 criteria for migraine, while 20 cases confirmed in the neurology clinic were included for the study with age matched healthy controls. DNA from blood samples were subjected for paired-end whole-exome sequencing. After quality control, 3365343 missense, frameshift, mis- sense splice region variants and insertion-deletion (indels) polymorphisms were tested for associ-ation with migraine. Significant variants were validated using Sanger sequencing. Seventeen 28
(p-value 9.091×10-05) functional variants in 12 genes (RETNLB, SCAI, ADH4, ESPL1, CPT2, FLG, PPP4R1, SERPINB5, ZNF66, ETAA1, EXO1 and CPA6) were associated with higher migraine risk including a stop gained frameshift (-13-14*SX) variant in the gene RETNLB (rs5851607; p-value 31
3.446×10-06). Gene analysis revealed that half of the significant novel migraine risk genes expressed in the temporal lobe (p-value 0.0058) of the cerebral cortex. The authors claim that this is the first study exploring the migraine risk 17 functional variants in 12 genes among Saudi female migraineurs of Arab ancestry using whole-exome sequencing. Half of the significant genes were expressed in the temporal lobe, which expands migraine pathophysiology and early identification using biomarkers for research possibilities on personalised genetics

General comment: Unfortunately the rebuttal letter is not satisfactory and some major (and big) point still remain. In particular, the authors did not provide a clear answer to the following points:

*) A big limitation of the value of this work is the small number of participants. Therefore the authors should clearly explain what could be the statistical significance of this analysis related to a so small group of people and the resulting limitations to their finding.

The authors response was: “The study was conducted during the pandemic (COVID-19) due to which the number of participants was limited”.

However, the limited number of participant results in a very limited statistical significance of the results. As a consequence, the authors should still show the value of the whole work.

*) The authors should clearly explain the statistical methods used in this work. The reference to the SPSS is not enough. In addition, they should explain in a detailed way what the classification tool “DAVID” is and how it works.

The authors response was” In the statistical analysis section, it is mentioned that SPSS version 21 is used for p-value calculation and DAVID is a classification tool used for identifying the significance of the site of expression. “

As a consequence, the authors do not provide (for the interested readers) a clear explanation of the used statistical methods. They should provide it in the next version of the revised manuscript.

Finally, even if a small conclusion section has been provided the authors should better present to overall value of their work, since it is not clear how this work could “open new opportunities for migraine pathophysiology and genetic research”. Indeed, the main big question about the overall statistical significance of the work still remain.

Author Response

*) The authors should clearly explain the statistical methods used in this work. The reference to the SPSS is not enough. In addition, they should explain in a detailed way what the classification tool “DAVID” is and how it works.

As per the suggestions by the respected reviewer the following details have been added in the revised version of the methodology 

“Student t-test was used for identifying the significant difference among the demographic characteristics between migraineurs and the controls.” 

Regarding “DAVID” we have replaced the old statement “The most significant top 10 genes were analysed using the gene functional classification tool DAVID to identify the significance in the site of expression [15]” with the following statement. 

“The most significant top 10 genes were analysed using the gene functional classification tool DAVID to identify the significance in the site of expression under the category of GNF_U133A_QUARTILE (p-value < 0.05) [15]” 

Finally, even if a small conclusion section has been provided the authors should better present to overall value of their work, since it is not clear how this work could “open new opportunities for migraine pathophysiology and genetic research”.

The following statement has been removed from the revised version of the manuscript. 

“The added migraine risk associated genes on the temporal lobe opens new opportunities for migraine pathophysiology and genetic research.” 

Reviewer 3 Report

The authors resolved my concerns and made great improvements in the revised manuscript. I accept the changes in the revision.

Author Response

We are thank full to the reviewer for the comments and accepting our changes to improve the manuscript.